# The Multikinase Inhibitor AD80 Induces Mitotic Catastrophe and Autophagy in Pancreatic Cancer Cells

**DOI:** 10.3390/cancers15153866

**Published:** 2023-07-29

**Authors:** Keli Lima, Lívia Bassani Lins de Miranda, Anali Del Milagro Bernabe Garnique, Bruna Oliveira de Almeida, Mariane Cristina do Nascimento, Guilherme Augusto Sousa Alcântara, Glaucia Maria Machado-Santelli, Eduardo Magalhães Rego, João Agostinho Machado-Neto

**Affiliations:** 1Laboratory of Medical Investigation in Pathogenesis and Targeted Therapy in Onco-Immuno-Hematology (LIM-31), Department of Internal Medicine, Hematology Division, Faculdade de Medicina, University of São Paulo, São Paulo 01246-903, Brazil; kelilima@usp.br (K.L.); marianenasc@hotmail.com (M.C.d.N.); edumrego@hotmail.com (E.M.R.); 2Department of Pharmacology, Institute of Biomedical Sciences, University of São Paulo, São Paulo 05508-000, Brazil; liviamirands@usp.br (L.B.L.d.M.); bruolialmeida@usp.br (B.O.d.A.); drguilhermealcantara.usp@gmail.com (G.A.S.A.); 3Department of Cell and Developmental Biology, Institute of Biomedical Sciences, University of São Paulo, São Paulo 05508-000, Brazil; anabega@usp.br (A.D.M.B.G.); glaucia.usp@gmail.com (G.M.M.-S.)

**Keywords:** pancreatic cancer, AD80, multikinase inhibitor, autophagy, aurora kinases

## Abstract

**Simple Summary:**

Pancreatic cancer is one of the most lethal human neoplasms, and its therapeutic repertoire remains limited. Advances in understanding the molecular complexity involved in the biology of the disease have paved the way for new therapeutic opportunities. AD80 is a multikinase inhibitor that inhibits S6K as well as RET, RAF, and SRC and displays antineoplastic effects in hematological and solid tumors. In the present study, we report the potential of AD80 as an antineoplastic agent for pancreatic cancer and the cellular and molecular changes induced by the drug.

**Abstract:**

Significant advances in understanding the molecular complexity of the development and progression of pancreatic cancer have been made, but this disease is still considered one of the most lethal human cancers and needs new therapeutic options. In the present study, the antineoplastic effects of AD80, a multikinase inhibitor, were investigated in models of pancreatic cancer. AD80 reduced cell viability and clonogenicity and induced polyploidy in pancreatic cancer cells. At the molecular level, AD80 reduced RPS6 and histone H3 phosphorylation and induced γH2AX and PARP1 cleavage. Additionally, the drug markedly decreased AURKA phosphorylation and expression. In PANC-1 cells, AD80 strongly induced autophagic flux (consumption of LC3B and SQSTM1/p62). AD80 modulated 32 out of 84 autophagy-related genes and was associated with vacuole organization, macroautophagy, response to starvation, cellular response to nitrogen levels, and cellular response to extracellular stimulus. In 3D pancreatic cancer models, AD80 also effectively reduced growth independent of anchorage and cell viability. In summary, AD80 induces mitotic aberrations, DNA damage, autophagy, and apoptosis in pancreatic cancer cells. Our exploratory study establishes novel targets underlying the antineoplastic activity of the drug and provides insights into the development of therapeutic strategies for this disease.

## 1. Introduction

Pancreatic cancer is one of the most lethal human malignancies presenting a 5-year survival rate of less than 10% [1], indicating that current therapeutic interventions for this cancer are far from satisfactory [2]. In recent years, significant advances in understanding the molecular complexity of the development and progression of pancreatic cancer opened the opportunity for new treatments [3]. Surgical resection is a curable therapy for pancreatic cancer patients with early stages of the disease, and gemcitabine (alone or in combination with target therapies) is a backbone drug for locally advanced and metastatic pancreatic cancer patients [2,4]. The use of systemic combination chemotherapies (i.e., 5-fluorouracil, folinic acid, irinotecan, and oxaliplatin, or/and gemcitabine plus nab-paclitaxel) has gained increasing prominence in the treatment of this type of tumor. These findings highlight the importance of drugs that act on multiple targets to exert their antineoplastic activity and prevent the emergence of resistant clones that clinically culminate in disease relapse and poor survival outcomes [5,6].

AD80 is a multikinase inhibitor optimized to perform RAF-ERK inhibition without acting directly on mTOR, which prevents a reactivation loop of the MAPK pathway [7]. Furthermore, it is well established now that AD80 prevents the phosphorylation of ribosomal protein S6 (RPS6), RET, p38 MAPK, and SRC, all proteins associated with cell proliferation and survival, in different cellular models [7,8,9,10,11]. Gain-of-function KRAS mutations that lead to constitutive activation of the MAPK pathway play a critical role in the initiation and maintenance of pancreatic cancer (frequency about 65%), highlighting this pathway as a pharmacological target in this disease [3,12].

The antineoplastic effects of AD80 have been reported in various solid tumors [8,10,13]. However, using monolayer or 3D cell culture models, we identified new potential molecular targets of AD80 in pancreatic cancer models and demonstrated a promising role in activating autophagy. Thus, the present study investigated the cellular and molecular mechanisms underlying the suppressive effects of AD80 on pancreatic cellular models.

## 2. Materials and Methods

### 2.1. Cell Culture and Reagent Chemicals

MIA PaCa-2 cells were kindly provided by Prof. Andrei Leitão (Institute of Chemistry of São Carlos, University of São Paulo, São Paulo, Brazil). PANC-1, AsPC-1, and HPDE cells were kindly provided by Prof. Daniela Sanchez Basseres (Institute of Chemistry, University of São Paulo, São Paulo, Brazil). Cell culture conditions were based on the recommendations of the American Type Culture Collection (ATCC). According to the Cellosaurus database (https://www.cellosaurus.org/, (accessed on 1 June 2023)), the driven mutations found are KRAS^G12C^ and TP53^R248W^ in MIA PaCa-2; KRAS^G12D^ and TP53^R273H^ in PANC-1; and KRAS^G12D^, CDKN2A^L78Hfs*41^, SMAD4^R100T^, and TP53^C135Afs*35^ in AsPC-1 cells. AD80 was purchased from TargetMol (Target Molecule Corp., Boston, MA, USA). Gemcitabine and bafilomycin A1 were obtained from Sigma-Aldrich (Sigma-Aldrich, St. Louis, MO, USA). Dimethyl sulfoxide (Me_2_SO_4_; DMSO) was obtained from Synth (Diadema, Brazil) and used as the drug dilution vehicle.

### 2.2. Cell Viability Assay

Cell viability was determined with a methylthiazoletetrazolium (MTT) assay. A total of 5 × 10^3^ cells per well were plated in a 96-well plate and exposed to vehicle or increasing concentrations of AD80 (0.032, 0.16, 0.8, 4, 20, and 100 µM) for 24, 48, or 72 h. For intermittent drug exposure assays, cells were exposed to vehicle or AD80 (0.032, 0.16, 0.8, 4, 20, and 100 µM) for 6, 12, 24, or 48 h, followed by incubation in drug-free media for 72 h. For combined treatment assays, cells were exposed to vehicle AD80 (0.25, 0.5, 1, 2, and 10 µM) and/or gemcitabine (0.62, 1.25, 2.5, 5, and 10 µM) for 72 h. Then, 10 μL MTT solution (5 mg/mL) was added, and the cells were incubated at 37 °C in 5% CO_2_ for 4 h. The reaction was stopped by adding 100 μL 0.1N HCl in anhydrous isopropanol. Cell viability was evaluated by measuring the absorbance at 570 nm. The IC_50_ values were calculated by performing a nonlinear regression analysis in GraphPad Prism 5 (GraphPad Software, Inc., San Diego, CA, USA). The selectivity index was calculated as the ratio of the IC_50_ for the non-tumor pancreatic cell line (HPDE) to the IC_50_ for pancreatic cancer cell lines.

### 2.3. Colony Formation Assay

Pancreatic cancer cell lines (1 × 10^3^ cells/35 mm plate) were incubated with vehicle or AD80 (0.016, 0.08, 0.4, 2, and 10 μM). Colonies were detected after 10–15 days of culture by staining with 0.5% crystal violet (Sigma-Aldrich) in 10% ethanol. Images were acquired using the G: BOX Chemi XRQ (Syngene, Cambridge, UK) and analyzed using the ImageJ software (US National Institutes of Health, Bethesda, MD, USA).

### 2.4. Cell Cycle Analysis

Pancreatic cancer cell lines were seeded in 60 mm cell culture dishes (2 × 10^5^ cells per plate) and cultured in the presence of a vehicle or AD80 (0.25, 0.5, and 1 μM) for 72 h. Then, cells were fixed with 70% ethanol for at least 4 h and incubated for 30 min with staining solution (0.1% Triton-X 100, 0.1 mg/mL of RNAse (Merck, Darmstadt, Germany) and 1 μg/mL of propidium iodide (Sigma-Aldrich). DNA content distribution was acquired in a FACSCalibur cytometer (Becton Dickinson, Lincoln Park, NJ, USA), and the data were analyzed using the FlowJo software v.X.0.7 (Treestar, Inc., San Carlos, CA, USA).

### 2.5. Morphology Analysis via Immunofluorescence

Pancreatic cancer cell lines treated with vehicle or 1 µM AD80 for 72 h were fixed with ice-cold 100% methanol, permeabilized with 0.5% Triton X-100 in PBS for 30 min at room temperature and blocked with 1% bovine serum albumin (BSA) in PBS for 1 h at room temperature. Next, the cells were incubated with anti-α-tubulin Alexa Fluor 488 conjugate (1:200 in 1% BSA in PBS; Thermo Fisher Scientific Inc., Cleveland, OH, USA) for 16 h at 4 °C protected from light, followed by washing once with PBS. Finally, the slides were mounted in ProLong Gold Antifade Mountant with DAPI (Thermo Fisher Scientific Inc.) for 1 h at room temperature. Images were captured using a fluorescent microscope (Lionheart FX Automated microscope; BioTek Instruments Inc., Santa Clara, CA, USA; magnification, 400×).

### 2.6. Cell Death Analysis

Cell death was determined using the propidium iodide (PI) and Hoechst 33342 (HO) double-staining method. Pancreatic cancer cell lines (2 × 10^5^ cells per well) were seeded in six-well plates and treated with vehicle or AD80 (0.25, 0.5, and 1 µM) for 72 h. Then, cells were washed with PBS and incubated with PI (5 µg/mL, Sigma-Aldrich) and HO (10 µg/mL, Sigma-Aldrich) and diluted in PBS for 20 min at 37 °C. Images were captured using a fluorescent microscope (Lionheart FX Automated microscope; BioTek Instruments Inc.; magnification, 100×). The percentage of dead cells was determined by calculating the fraction of PI-stained or fragmented nucleus cells relative to all cells. At least 2000 cells were scored in each experiment using the ImageJ software.

### 2.7. Western Blotting

Total protein extraction was performed using a buffer containing 100 mM Tris (pH 7.6), 1% Triton X-100, 2 mM PMSF, 10 mM Na_3_VO_4_, 100 mM NaF, 10 mM Na_4_P_2_O_7_, and 4 mM EDTA from pancreatic cancer cell lines treated with vehicle or AD80 (0.25, 0.5, or 1 µM) for 72 h. Alternatively, PANC-1 cells were treated with vehicle, AD80 (1 µM) and/or bafilomycin A1 (10 nM) for 72 h. Extracted proteins were quantified using the Bradford method. Equal amounts of protein (30 µg) were separated by 8–15% SDS-PAGE and transferred to nitrocellulose membranes (GE Healthcare, Milwaukee, WI, USA). Antibodies against phosphorylated(p)-RPS6 (#4858), RPS6 (#2217), p-ERK1/2 (#9101), ERK1/2 (#9102), p-AURKA/B/C (#2914), AURKA (#14475), p-histone H3^S10^ (#9701), histone H3 (#4499), SQSTM1/p62 (#88588), LC3BI/II (#2775), PARP1 (#9542), p-histone H2A.X^S139^ (γH2AX; #9718), and α-tubulin (#2144) were all obtained from Cell Signaling Technology, Inc (Beverly, MA, USA). All primary antibodies were diluted 1:2000 and incubated for 16 h at 4 °C. The α-tubulin protein was used as the loading control. Secondary anti-rabbit (#7074) and anti-mouse (#7076) antibodies conjugated with horseradish peroxidase were obtained from Cell Signaling Technology, Inc. and used at 1:2000 dilution with 2 h of incubation at room temperature. The SuperSignal West Dura Extended Duration Substrate system (Thermo Fisher Scientific, Inc.) and G: BOX Chemi XX6 gel document system (Syngene) were used for blot visualization. Cropped gels retain important bands, but whole gel images are available in Appendix A.

### 2.8. Acidic Vesicular Organelle Analysis via Fluorescence Microscopy

Pancreatic cancer cell lines treated with vehicle or 1 µM AD80 for 72 h were washed with PBS, resuspended in PBS containing 0.1 μg/mL acridine orange (Sigma-Aldrich), incubated for 30 min, and evaluated for GFP and RFP channels using a fluorescent microscope (Lionheart FX Automated microscope; Agilent BioTek Instruments Inc., Santa Clara, CA, USA; magnification, 200×).

### 2.9. PCR Array Analysis

Total RNA from PANC-1 cells treated with vehicle or AD80 (1 μM) for 72 h was obtained using TRIzol reagent (Thermo Fisher Scientific). The cDNA was synthesized from 2 μg RNA using an RT^2^ First Strand Kit (Qiagen Sciences Inc., Germantown, MD, USA). PCR array analysis was performed using a Human Autophagy RT^2^ Profiler PCR Array kit (#PAHS-084ZA; Qiagen) according to the manufacturer’s instructions. The mRNA levels were normalized to those in vehicle-treated cells, and genes that presented *p* < 0.05 (Student’s *t*-test) were included in the heatmap using Multiple Experiment Viewer (MeV) 4.9.0 software [14]. Amplification was performed using QuantStudio 3 Real-Time PCR System (Thermo Fisher Scientific, Inc.). A network for AD80 modulated genes was constructed using the GeneMANIA database (https://genemania.org/, (accessed on 1 June 2023)).

### 2.10. Quantitative PCR

Total RNA from MIA PaCa-2, PANC-1, and AsPC-1 cells treated with vehicle or AD80 (1 μM) for 72 h was obtained using TRIzol reagent (Thermo Fisher Scientific, Inc.). cDNA was synthesized from 1 µg RNA using a High-Capacity cDNA Reverse Transcription Kit (Thermo Fisher Scientific, Inc.). Quantitative PCR (qPCR) was performed using a QuantStudio 3 Real-Time PCR System (Thermo Fisher Scientific, Inc.), a SybrGreen System, and specific primers (Appendix A). *HPRT1* and *ACTB* were used as reference genes. Relative quantification values were calculated using the 2^−ΔΔCT^ equation [15]. A negative ‘No Template Control’ was included for each primer pair.

### 2.11. Soft Agar Assay

In a 12-well plate, 500 µL of 0.5% agarose was added to form the bottom layer and 500 µL was used for the upper layer, which contained the pancreatic cancer cells with 0.3% agarose. A total of 200 µL of the medium was added to the surface to prevent drying, and plates were kept for approximately 21 days until the visualization of colony formation. Then, plates were stained with MTT (5 mg/mL) for 45 min. Images were acquired using a G: BOX Chemi XRQ (Syngene) and analyzed using ImageJ v.1.45s software.

### 2.12. Cell Viability Analysis in the Spheroid Model

Pancreatic cancer cell lines (MIA PaCa-2: 2.5 × 10^3^ cells/100 μL, PANC-1: 1 × 10^4^ cells/100 μL, AsPC-1: 1 × 10^4^ cells/100 μL) were seeded per well on 96-well plates prepared with 65 μL 1% agarose on the bottom [15]. Spheroids were formed upon incubation at 37 °C and 5% CO_2_ for four days. Then, cells were treated with vehicle or AD80 (0.4, 2, and 10 μM) for 72 h. The images were obtained using an inverted digital microscope EVOS AME-3302. The viability of the spheroids was assessed with ATP quantification utilizing the CellTiter-Glo 3D Cell Viability kit (Promega, Madison, WI, USA) according to the manufacturer’s instructions.

### 2.13. Statistical Analysis

Statistical analyses were performed using GraphPad Prism 8 (GraphPad Software, Inc.). ANOVA followed by the post hoc Bonferroni’s test was used for multiple comparisons. The paired Student’s *t*-test was used to compare the two groups. Results with *p*-values of <0.05 were considered statistically significant.

## 3. Results

### 3.1. AD80 Exhibits Antineoplastic Activity in Pancreatic Cancer Cells

First, the effects of AD80 on viability were investigated in pancreatic cancer cells. As shown in Figure 1A, AD80 reduced cell viability in a concentration- and time-dependent manner, with MIA PaCa-2 cells being the most sensitive to the drug. The IC_50_ values ranged from 0.08 to 12.3 µM for MIA PaCa-2, 4.46 to 30.38 µM for PANC-1, and 0.33 to 43.24 µM for AsPC-1. In HPDE cells, IC_50_ values ranged from 1.48 to 4.78 µM and showed a favorable selectivity index (SI > 1) for MIA PaCa-2 and AsPC-1 cells after the 24 and 48 h treatments (Appendix A). Similarly, clonal growth was strongly inhibited through exposure to AD80 in all pancreatic cell lines tested (*p* < 0.05, Figure 1B,C). Next, an intermittent exposure assay was performed to better elucidate the role of drug exposure time in this context. A 6 h exposure to AD80 is enough to impact cell viability negatively, but the longer the exposure time, the more potent AD80 is at reducing cell viability (Figure 1D).

Next, the cellular events involved in the reduction of viability were investigated. AD80 treatment increased the cell population in the G_2_/M phases of the cell cycle and led to the appearance of a polyploidy population (>4 N) (all *p* < 0.05), which was most evident in PANC-1 cells (Figure 2A). An increase in subG1 cells was also observed, suggesting induction of apoptosis (*p* < 0.05, Figure 2A). The morphological analysis confirmed the increase in nuclear size and volume of MIA PaCa-2, PANC-1, and AsPC-1 cells (Figure 2B). Notably, in PANC-1 cells, it was possible to observe the presence of multiple nuclei, which confirms the most prominent findings in the assay for evaluating DNA content (Figure 2B). In addition, the number of apoptotic cells increased after AD80 exposure, corroborating the subG_1_ data from the flow cytometry findings (Figure 2C). Together these findings suggest that AD80 induces mitotic aberrations and cell death in pancreatic cancer cells.

### 3.2. AD80 Induces Mitotic Catastrophe and Autophagy Molecular Marker Expression in Pancreatic Cancer Cells

The effects of AD80 on the activation of proteins involved in proliferation, cell cycle progression, autophagy, apoptosis, and DNA damage were evaluated. AD80 reduced RPS6 activation, an expected on-target effect since the drug directly inhibits S6K [7]. The effects on ERK1/2 phosphorylation varied widely, from inhibition in AsPC-1 cells to activation in PANC1 cells. AD80 markedly reduced AURKA/B/C phosphorylation and AURKA expression (Figure 3A). PARP1 cleavage and γHA2X expression were observed in all cell lines but were more intense in MIA PaCa-2 cells, corroborating the greater sensitivity of this cell line to drug-induced cell death (Figure 3A). Confirming the presence of mitotic aberrations and increased γHA2X, increased levels of CHK1 and/or CHK2 phosphorylation, as well as the induction of genes involved in the response to DNA damage, were observed in pancreatic cells upon AD80 exposure (Appendix A). In PANC-1 cells, AD80 drastically reduced the expression of SQSTM1/p62 and LC3B indicating intense autophagic flux. In MIA PaCa-2 cells but not in AsPC-1 cells, a reduction in SQSTM1/p62 was also observed (Figure 3A).

Considering that the relationship between AD80 and autophagy is a little-explored topic, we deepen the investigations in this context. Acridine orange staining suggests an increase in acidic vesicular organelles (AVOs) in MIA PaCa-2 and PANC-1 cells but not in ASPC-1 cells, corroborating the molecular findings (Figure 3B). In PANC-1 cells, treatment with bafilomycin A1, a potent autophagic flux inhibitor, prevented the consumption of LC3B and SQSTM1 triggered by AD80, corroborating the hypothesis that AD80 is an autophagy inducer (Appendix A). Using a PCR array for autophagy-related genes, we observed that 32 out of 84 genes were modulated (28 upregulated and 4 downregulated) after exposure of PANC-1 cells to AD80 (all *p* < 0.05, Figure 3C and Appendix A). An integrated analysis of the modulated genes (fold-change > 1.5, *p* < 0.05) indicates that AD80 impacts several stages of autophagy, from sensitization to nutrient deprivation and vacuole organization to autophagy (all FDR q < 0.05, Figure 3D).

### 3.3. AD80 Reduces Anchorage-Independent Growth and Spheroid Cell Viability in Pancreatic Cancer

We performed anchorage-independent growth and spheroid generation assays to provide evidence of the antineoplastic potential of AD80 in more complex conditions. The anchorage-independent growth assay mimics the stress that the cells that leave the primary tumor go through to generate distal metastases, with most cells spontaneously entering a process of detachment-induced cell death known as anoikis [16,17]. Cells that survive this process are usually resistant to cell death. Interestingly, AD80 significantly inhibited anchorage-independent growth in all pancreatic cell lines tested (all *p* < 0.05, Figure 4A). On the other hand, the 3D model mimics an already-formed tumor, with cellular communications, nutrients gradient, and spheroid architecture, establishing a greater complexity compared to monolayer cellular models [18]. In this context, AD80 reduced the cell viability of pancreatic cancer spheroid models (all *p* < 0.05, Figure 4B). These results suggest that AD80 could treat primary tumors and attenuate their ability to spread to other sites.

### 3.4. AD80 Potentiates Gemcitabine-Reduced Cell Viability in PANC-1 and AsPC-1 Cells

Lastly, the effects of AD80 combined with gemcitabine, a frontline therapy for pancreatic cancer, were investigated in cellular models of the disease. In PANC-1 and AsPC-1 cells, but not MIA PaCa-2 cells, AD80 exposure potentiated the antineoplastic effects of gemcitabine on the reduction of cell viability (*p* < 0.05, Figure 5).

## 4. Discussion

Herein, we have investigated the effects of AD80 on malignant phenotype in monolayer and 3D pancreatic cancer cellular models. Remarkable antineoplastic effects, including the reduction of cell viability, clonal growth, cell cycle progression, and anchorage-free growth, were observed, in addition to the induction of cell death, autophagy, and mitotic aberrations. AD80 acts as a multikinase inhibitor and exerts antineoplastic effects in a variety of cancer types, including lymphoma [8], glioma [8], lung cancer [11], hepatocellular carcinoma [10], ovarian cancer [13], prostate cancer [19], acute leukemia [9], and colorectal cancer [20]. Although reduced cell viability is a similar aspect between different types of cancer, the versatility of AD80 allows it to act on various molecular targets, including S6K, RET, RAF, and p38 MAPK [7,8,9,10,11], making it an attractive multitarget compound. The use of combination therapies or drugs that act on multiple targets in antineoplastic therapy has been proposed to avoid the emergence of chemoresistant clones and subsequent refractoriness or disease relapse [5,6]. In our study, a 6 h AD80 exposure generated irreversible data that reflected in the reduction of long-term cell viability, indicating the high sensitivity of this type of cancer to the drug.

In the present study, aurora kinases (AURKs) were downregulated with AD80 in pancreatic cancer cells. The relevant role of AURKs in this disease has been described in previous studies, especially for AURKA. AURKA expression is stimulated via a KRAS mutation, and high AURKA expression predicts unfavorable clinical outcomes in patients with pancreatic cancer. AURKA inhibition with siRNA or pharmacological inhibitors reduces the in vitro and in vivo proliferation, clonogenicity, and survival of MIA PaCa-2, PANC-1, and/or AsPC-1 cells [21,22]. The inhibition of AURKs via AD80 was reported in the initial characterization of the drug. In vitro, the drug reduces AURKA, AURKB, and AURKC activities by 76%, 87%, and 58%, respectively [7]. AURKs are essential proteins for the correct segregation of chromosomes and the success of mitosis and cytokinesis [23,24]. Thus, the cellular effects noticed in pancreatic cancer models upon AD80 exposure indicate that AURKs may be one of the main targets associated with the observed phenotype.

AD80-induced autophagic flow activation was a breakthrough finding in our study. mTOR-mediated signaling is one of the main physiological inhibitors of autophagy [25]. AD80 was specifically designed not to directly inhibit mTOR, preventing the reactivation of the mTOR/RAF/ERK regulatory loop, but the drug inhibits S6K, a critical downstream target of mTOR [7]. In our study, the induction of autophagy in pancreatic cancer cells was independent of the inhibition of RPS6 phosphorylation status, indicating that other molecular mechanisms may be involved. Our exploratory analysis based on gene expression suggests that AD80 alters the expression of genes involved in multiple stages of autophagy, from sensitization through nutrient deprivation to the organization of the autophagic vacuole.

Our study opens perspectives for novel antineoplastic agents in pancreatic cancer; however, it has limitations. The scope of the present study was based on in vitro studies using pancreatic cancer cell lines, which provided cellular and molecular information. Investigating the effects in animal models of pancreatic cancer (e.g., ectopic or orthotropic tumor models) is needed to define the selectivity and toxicity of the drug. The results could lead to clinical trials.

## 5. Conclusions

In summary, our results indicate that AD80 exerts multiple antineoplastic effects in monolayer and 3D pancreatic cancer cellular models, leading to mitotic catastrophe and autophagy. Our preclinical findings highlight AD80 as a multitarget drug that may contribute to the therapeutic repertoire against pancreatic cancer.

## Figures and Tables

**Figure 1 cancers-15-03866-f001:**
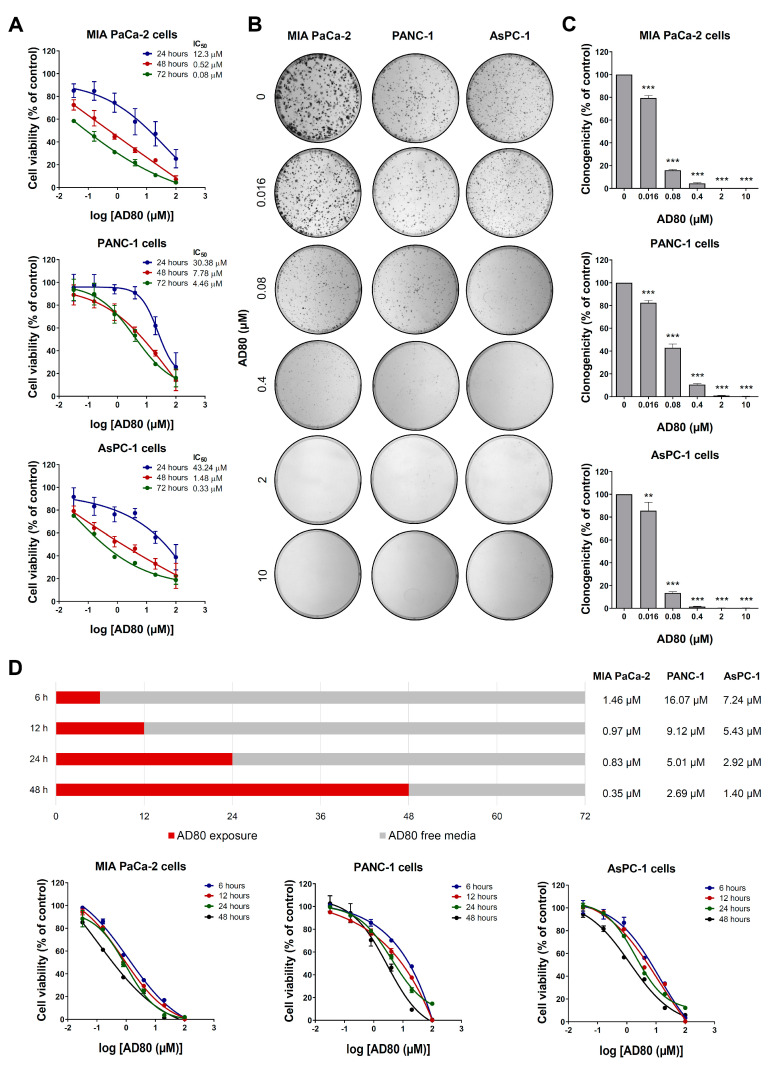
AD80 reduces pancreatic cancer cell viability and clonogenicity. (**A**) Dose- and time-response cytotoxicity was evaluated with the MTT assay. MIA PaCa-2, PANC-1, and AsPC-1 cells were treated with vehicle or increasing concentrations of AD80 (0.032, 0.16, 0.8, 4, 20, and 100 µM) for 24, 48, and 72 h. Values are expressed as the percentage of viable cells for each condition relative to vehicle-treated cells. Results are shown as the mean ± SD of at least three independent experiments. (**B**) Colony formation images from pancreatic cancer cells treated with vehicle or AD80 (0.016, 0.08, 0.4, 2, and 10 μM) for 10–15 days. (**C**) The bar graph represents the mean ± SD of the relative number of colonies (% of control). ** *p* < 0.01, *** *p* < 0.001; ANOVA and Bonferroni post-test. (**D**) The bar graph indicates the duration of treatment with AD80 (red bar) and the incubation time in a drug-free medium (grey bar) for the intermittent treatment in MIA PaCa-2, PANC-1, and AsPC-1 cells. The IC_50_ values are described in the Figure. Dose–response cytotoxicity curves are illustrated for each condition. Results are shown as the mean ± SD of at least three independent experiments.

**Figure 2 cancers-15-03866-f002:**
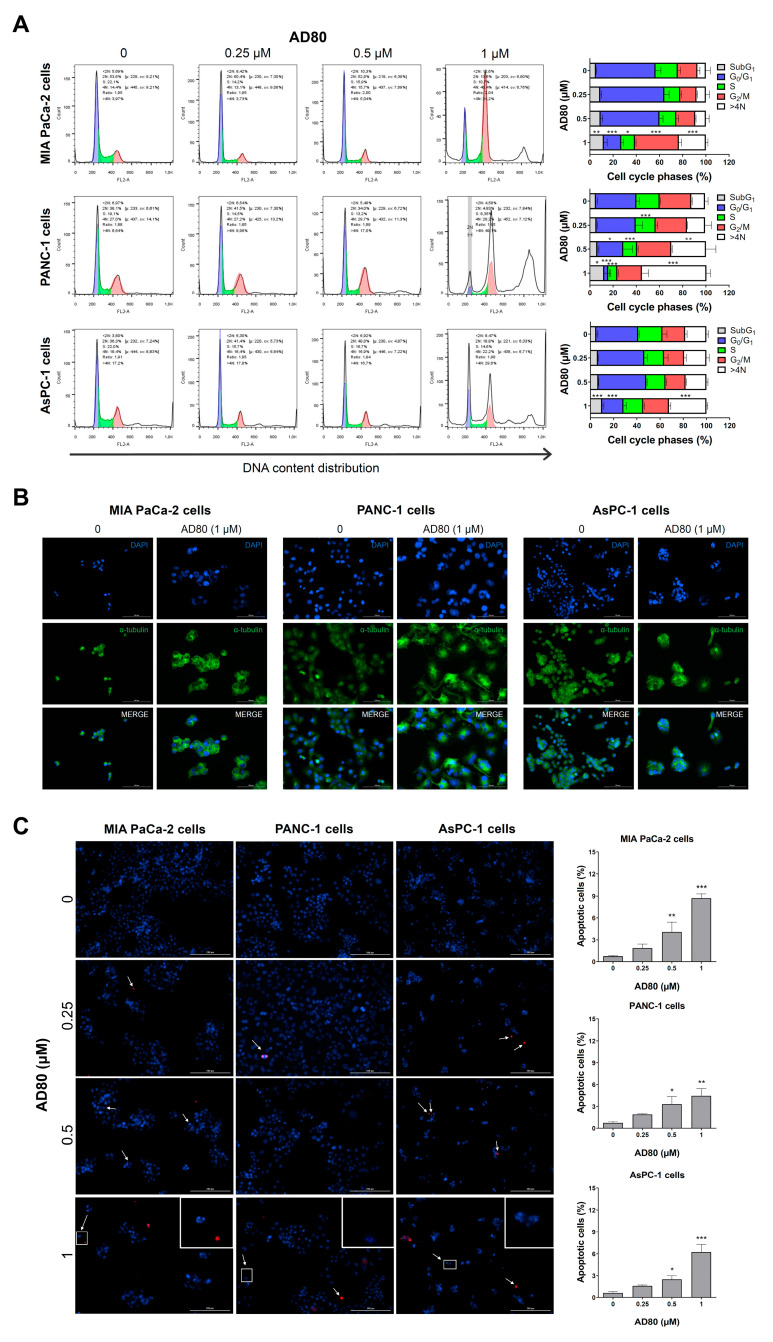
AD80 induces mitotic aberrations, polyploidy, and cell death in pancreatic cancer cells. (**A**) DNA content distribution was evaluated by staining with propidium iodide and flow cytometry in MIA PaCa-2, PANC-1, and AsPC-1 cells treated with vehicle or AD80 (0.25, 0.5, and 1 µM) for 72 h. A representative histogram for each condition is illustrated. The mean ± SD of at least three independent experiments is represented in the bar graph. The *p*-values are indicated; * *p* < 0.05, ** *p* < 0.01, *** *p* < 0.001; ANOVA and Bonferroni post-test. (**B**) Immunofluorescence analysis of pancreatic cancer cells treated with 1 µM AD80 or vehicle for 72 h, displaying α-tubulin (green) and DAPI (blue) staining. Scale bar, 100 µm. (**C**) Representative images displaying nuclear staining for Hoechst 33342 (HO, blue) and propidium iodide (PI, red) in MIA PaCa-2, PANC-1, and AsPC-1 cells treated with vehicle or AD80 (0.25, 0.5, and 1 µM) for 72 h. Scale bar, 200 µm. The apoptosis rate was expressed as the percentage of PI-positive plus fragmented nucleus cells relative to all HO-positive cells (at least 200 cells were scored per condition/experiment). The bar graph represents the mean ± SD of at least three independent experiments. * *p* < 0.05, ** *p* < 0.01, *** *p* < 0.001; ANOVA and Bonferroni post-test.

**Figure 3 cancers-15-03866-f003:**
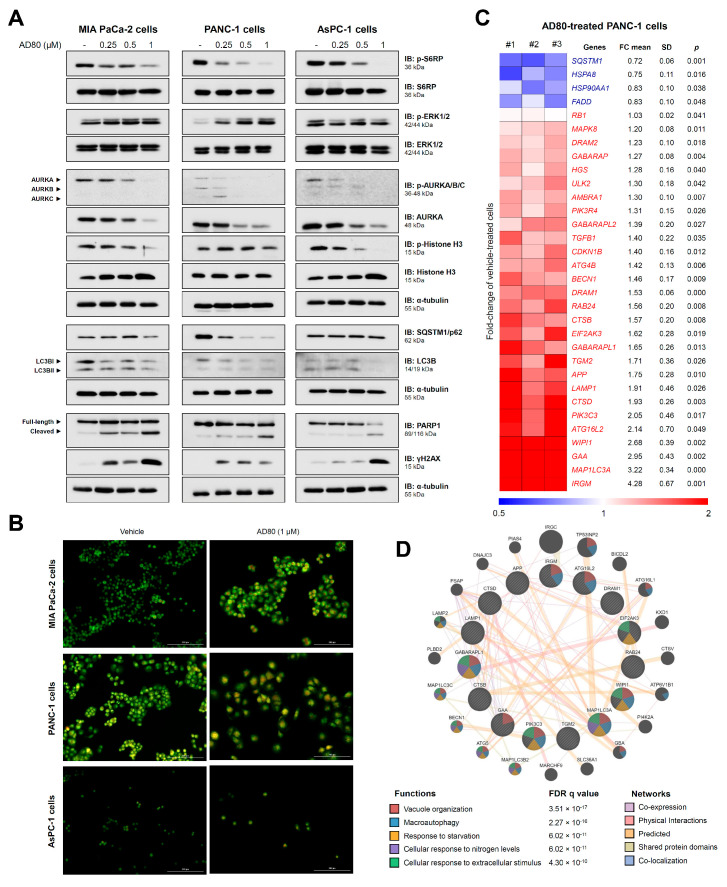
AD80 triggers mitotic catastrophe and autophagy molecular markers in pancreatic cancer cells. (**A**) Western blot analysis of phosphorylated(p)-RPS6, p-ERK1/2, p-AURKA/B/C, p-histone H3, SQSTM1/p62, LC3BI/II, γH2AX, and PARP1 (total and cleaved) in whole cell extracts from MIA PaCa-2, PANC-1, and AsPC-1 cells treated with vehicle or AD80 (0.25, 0.5, and 1 µM) for 72 h. Membranes were reprobed with the antibody to detect total protein and/or α-tubulin. (**B**) The evaluation of acidic vesicular organelles was investigated through acridine orange labeling and fluorescence microscopy in pancreatic cell lines treated with vehicle or AD80 (1 μM) for 72 h. The overlapping GFP and RFP channels are shown. Scale bar, 100 µm. (**C**) Autophagy-related genes significantly modulated in AD80- (1 μM) compared to vehicle-treated PANC-1 cells were included in the heatmap (n = 3). Blue indicates decreased mRNA levels, and red indicates increased mRNA levels. The fold-change (FC) mean, standard deviation (SD), and *p*-value (Student’s t-test) are shown in the figure. (**D**) A network for AD80-modulated genes (FC > 1.5 in either direction) was constructed using the GeneMANIA database (https://genemania.org/, (accessed on 1 June 2023)). Significantly modulated genes are illustrated as crosshatched circles, and circles without crosshatching indicate interacting genes. The main biological interactions and associated functions are described in the figure. FDR: false discovery rate. Note that functions related to all stages of autophagy were found.

**Figure 4 cancers-15-03866-f004:**
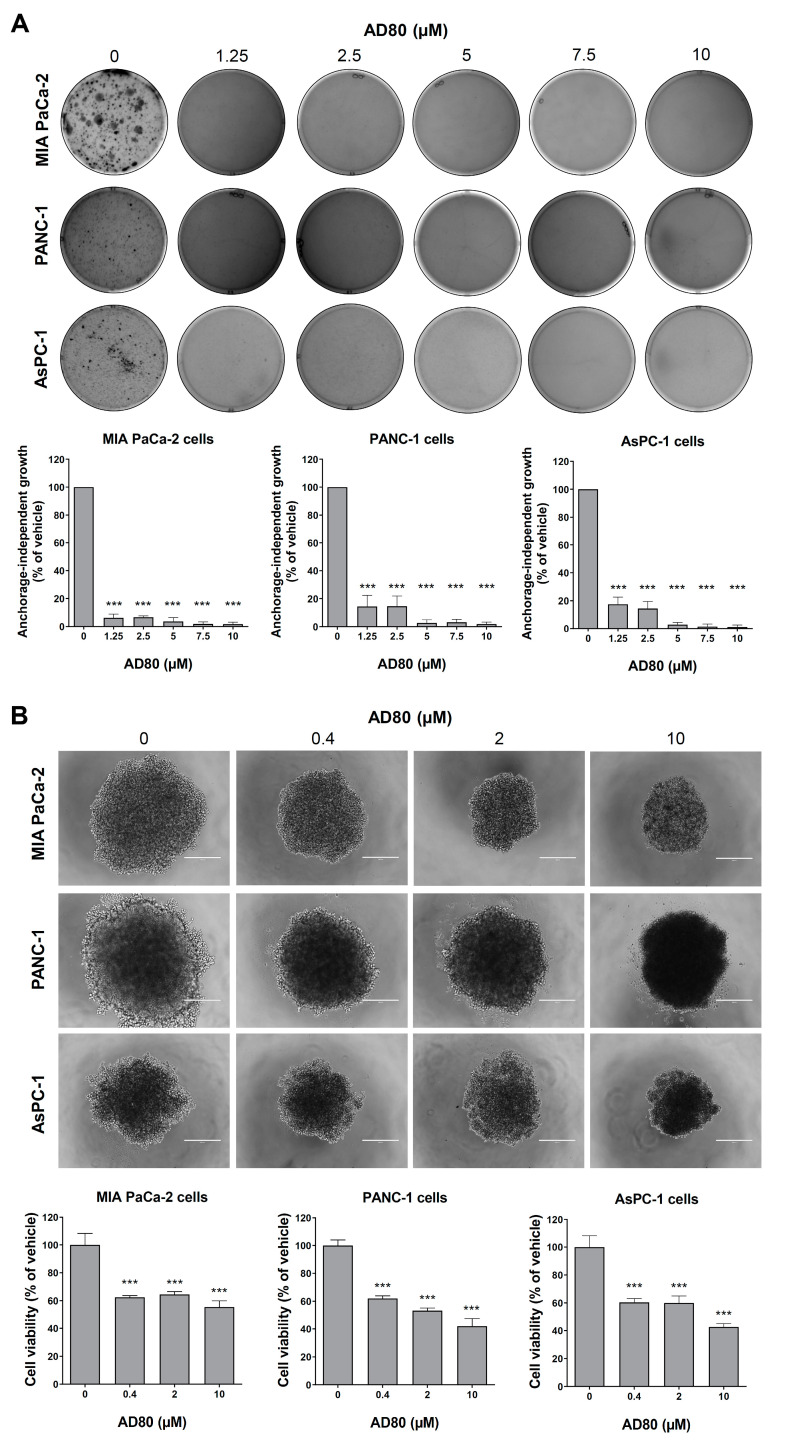
Exposure to AD80 decreases anchorage-independent growth and spheroid cell viability in pancreatic cancer models. (**A**) Anchorage-independent growth images from pancreatic cancer cells treated with vehicle or AD80 (1.5, 2.5, 5, 7.5, and 10 μM) for 21 days. The bar graph represents the mean ± SD of the relative number of colonies (% of the vehicle). *** *p* < 0.001; ANOVA and Bonferroni post-test. (**B**) Representative spheroid images from pancreatic cancer cells treated with vehicle or AD80 (0.4, 2, and 10 μM) for 72 h. The bar graph represents the mean ± SD of the spheroid viability (% of the vehicle). Scale bar, 100 µm. *** *p* < 0.001; ANOVA and Bonferroni post-test.

**Figure 5 cancers-15-03866-f005:**
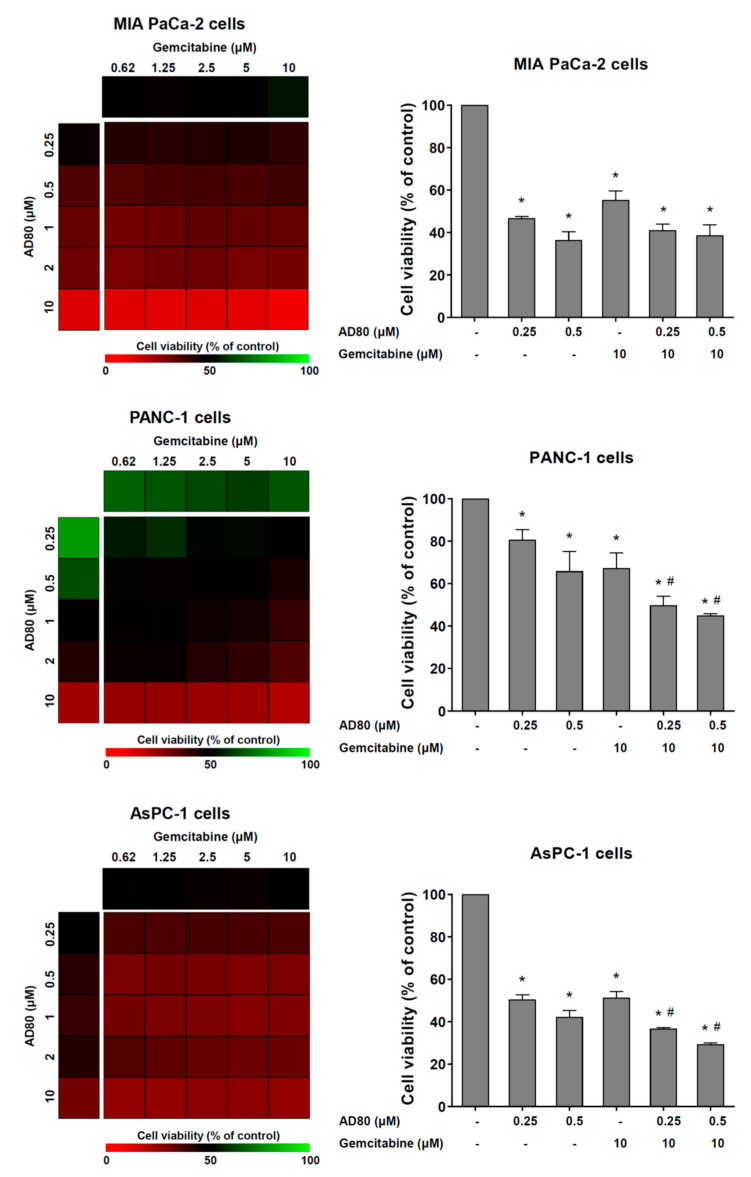
AD80 potentiates gemcitabine-reduced cell viability in PANC-1 and AsPC-1 cells. Dose–response cytotoxicity for combined treatment was analyzed using an MTT assay for MIA PaCa-2, PANC-1, and AsPC-1 cells treated with vehicle or graded concentrations of AD80 (0.25, 0.5, 1, 2, and 10 µM) and gemcitabine (0.62, 1.25, 2.5, 5, and 10 µM) alone or in combination with each other for 72 h, as indicated. Values are expressed as the percentage of viable cells for each condition relative to vehicle-treated cells. Results are shown as the mean of at least three independent experiments. Bar graphs represent the cell viability for selected concentrations. The *p*-values and cell lines are indicated; * *p* < 0.05 treatment versus vehicle, # *p* < 0.05 monotherapy versus combined therapy; ANOVA test and Bonferroni post-test.

## Data Availability

The datasets used and/or analyzed during the current study are available from the corresponding author upon reasonable request.

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
