# Peer review of "The Multikinase Inhibitor AD80 Induces Mitotic Catastrophe and Autophagy in Pancreatic Cancer Cells"

_cancers, 2023, doi:10.3390/cancers15153866_

Round 1

Reviewer 1 Report

Cancers (ISSN 2072-6694)

Manuscript ID: cancers-2467205

The manuscript titled " The Multikinase Inhibitor AD80 Induces Mitotic Catastrophe 2 and Autophagy in Pancreatic Cancer Cells " by Lima et al. is interesting and provides new therapeutic insight for Pancreatic Cancer patients. However, AD80 is widely studied in several cancers, recently well-studied in colorectal (PMID: 36030982) and ovarian cancer (PMID: 33488949). This study investigates the impact of AD80 on pancreatic cancer cells, revealing its ability to induce mitotic abnormalities, DNA damage, autophagy, and apoptosis. Additionally, the research elucidates the specific cellular and molecular mechanisms underlying the suppressive effects of AD80 on pancreatic cellular models (2D or 3D/spheroids culture system). However, the current study requires further refinement and additional experimental analysis to provide direct evidence that supports the hypothesis and substantiates the findings.

1.     An important point. The author should explain why MIA-PaCa-2, PANC-1and AsPC1 cells have been selected throughout the study. The KRAS status of each cell line should also be mentioned. Does KRAS mutation modify AD80 response?

2.     To enhance the comprehensiveness of the study, it would be advantageous for the author to incorporate a normal pancreatic cell line, such as HPDE or immortalized HPNE, in order to illustrate the effects of AD80 on non-cancerous cells.

3.     The authors noted a discrepancy in the response to AD80 across the three cell lines used in each set of experiments. It is crucial to thoroughly discuss the possible reasons behind this observed variation.

 4.     Pancreatic cancer is associated with extreme drug resistance. Does long-term treatment with AD80 lead to drug resistance?

 5.     Another major issue. In vitro AD80 treatment at µM doses is beyond physiological doses. In vivo evaluation of AD80 toxicity and AD80 anticancer effect using an ectopic or orthotropic PDAC mouse model would have been extremely valuable.

 6.     The authors should devote more effort to enhancing the discussion section of their manuscript, focusing on effectively highlighting the novel insights derived from their study and providing valuable perspectives for future investigations and therapeutic applications.

Minor editing of English language required.

Author Response

Reviewer #1

The manuscript titled "The Multikinase Inhibitor AD80 Induces Mitotic Catastrophe 2 and Autophagy in Pancreatic Cancer Cells" by Lima et al. is interesting and provides new therapeutic insight for Pancreatic Cancer patients. However, AD80 is widely studied in several cancers, recently well-studied in colorectal (PMID: 36030982) and ovarian cancer (PMID: 33488949). This study investigates the impact of AD80 on pancreatic cancer cells, revealing its ability to induce mitotic abnormalities, DNA damage, autophagy, and apoptosis. Additionally, the research elucidates the specific cellular and molecular mechanisms underlying the suppressive effects of AD80 on pancreatic cellular models (2D or 3D/spheroids culture system). However, the current study requires further refinement and additional experimental analysis to provide direct evidence that supports the hypothesis and substantiates the findings.

1. An important point. The author should explain why MIA-PaCa-2, PANC-1and AsPC1 cells have been selected throughout the study. The KRAS status of each cell line should also be mentioned. Does KRAS mutation modify AD80 response?

Authors' response: The authors thank the reviewer for allowing us to improve our manuscript. The cell lines used in the study have different profiles in terms of morphology, adhesion, and cell growth rate, which could represent the heterogeneity observed in the clinic with pancreatic cancer patients. Following the reviewer's suggestion, we have added the mutational status of the main driven-mutation cell lines in the Material and Methods section. All cell lines included in the study have mutations in KRAS and TP53; thus, it is impossible to make a direct association between KRAS mutational status and drug response.

The following sentence has been included in the revised version of the manuscript:

“According to the Cellosaurus database (https://www.cellosaurus.org/), driven-mutations found are KRASG12C and TP53R248W in MIA PaCa-2; KRASG12D and TP53R273H in PANC-1; and KRASG12D, CDKN2AL78Hfs*41, SMAD4R100T, and TP53C135Afs*35 in AsPC-1.”

2. To enhance the comprehensiveness of the study, it would be advantageous for the author to incorporate a normal pancreatic cell line, such as HPDE or immortalized HPNE, in order to illustrate the effects of AD80 on non-cancerous cells.

Authors' response: The authors thank the reviewer for the suggestion and opportunity to improve the presentation of our results. Following the reviewer's suggestion, we have added cell viability experiments in HPDE cells (non-tumor pancreatic cells) in the current version of the manuscript.

The following sentence has been included in the revised version of the manuscript:

"In HPDE cells, IC50 values ranged from 1.48 to 4.78 µM and showed a favorable selectivity index (SI > 1) for MIA PaCa-2 and AsPC-1 cells after the 24 and 48 h treatments (Supplementary Figure 2)."

3. The authors noted a discrepancy in the response to AD80 across the three cell lines used in each set of experiments. It is crucial to thoroughly discuss the possible reasons behind this observed variation.

Authors' response: As previously mentioned, the cell lines included in the study present mutations in KRAS and TP53, and it is impossible to make a direct association between the mutational status of KRAS and the response to the drug. In addition, there are other differences concerning morphology, adherence and growth rate, which reflects the clinically observed heterogeneity in patients with pancreatic cancer. Thus, we could not establish a causal relationship between the response to the drug and the molecular profile, and any comments in this context are merely speculative.

4. Pancreatic cancer is associated with extreme drug resistance. Does long-term treatment with AD80 lead to drug resistance?

Authors' response: In the colony formation and anchorage-free growth assays, the cells were exposed to the drug for 10 to 21 days, and the appearance of resistant clones was rare at concentrations greater than 2 µM.

5. Another major issue. In vitro AD80 treatment at µM doses is beyond physiological doses. In vivo evaluation of AD80 toxicity and AD80 anticancer effect using an ectopic or orthotropic PDAC mouse model would have been extremely valuable.

Authors' response: The authors thank the reviewer for the valuable suggestion and the opportunity to clarify this topic. Our research group is initiating pancreatic cancer studies, and unfortunately, we do not have standardized murine models for pancreatic cancer in our laboratory, which will take some time. However, the most significant limitation at the moment is the lack of research funding needed to initiate in vivo studies of AD80. We hope that the reviewer understands our current difficulties. We acknowledge this limitation in the discussion of the revised version of the manuscript.

The following sentences have been included in the revised version of the manuscript:

" Our study opens perspectives for novel antineoplastic agents in pancreatic cancer; however, it has limitations. The scope of the present study was based on in vitro studies using pancreatic cancer cell lines, which provided cellular and molecular information. Investigating the effects in animal models of pancreatic cancer (e.g., ectopic or orthotropic tumor models) is needed to define the selectivity and toxicity of the drug.”

6. The authors should devote more effort to enhancing the discussion section of their manuscript, focusing on effectively highlighting the novel insights derived from their study and providing valuable perspectives for future investigations and therapeutic applications.

Authors' response: The authors thank the reviewer for the opportunity to improve the discussion of our manuscript.

The following sentences have been included in the revised version of the manuscript:

"Herein, we have investigated the effects of AD80 on malignant phenotype in mono-layer and 3D pancreatic cancer cellular models. Remarkable antineoplastic effects, including reduction of cell viability, clonal growth, cell cycle progression, and anchor-age-free growth, were observed, in addition to the induction of cell death, autophagy, and mitotic aberrations."

Reviewer 2 Report

The article by Lima et al evaluated the efficacy of a multikinase inhibitor AD80 in pancreatic cancer in vitro. They had shown that AD80 lowered cell viability and clonogenicity and increased polyploidy in pancreatic cancer cells in vitro. They also performed Western blot and imaging studies to understand the molecular mechanism. Not surprisingly, AD80 showed alterations in multiple signaling pathways due to its role as a multikinase inhibitor. A targeted gene expression analysis by PCR array showed that AD80 could modulate a significant portion of autophagy-related genes and pathway analyses revealed its association with vacuole organization, macroautophagy, response to starvation, and cellular response to extracellular stimulus among others. Finally, the efficacy of AD80 in inhibiting growth in anchorage-independent soft agar assay as well as in 3D spheroid models was determined, albeit at higher dose levels than previous experiments. However, the manuscript needs significant improvement.

1.       First and foremost, no in vivo xenograft studies were performed to determine the efficacy of AD80 as a viable therapeutic in pancreatic cancer. Multikinase inhibitors typically show very good in vitro efficacy due to their multi-target inhibition, however, they might cause significant toxicity at therapeutic doses in vivo due to the same reason. Hence, an in vivo experiment is needed to nullify any potential toxicity issues. This is a serious limitation of this study.

2.       The effect of AD80 on non-cancer cells such as HPDE cells should also be tested.

3.       Please include cell cycle checkpoint markers ATM/CHk2, and ATR/CHK1 in western blot experiments. This is important since mitotic catastrophe is typically caused by the deregulation of cell cycle checkpoints.

4.       While p62 is considered an autophagic flux marker, alternative experiments like comparing LC3B conversion in the presence and absence of a lysosomal inhibitor such as bafilomycin-A1 should be performed to confirm the flux.

5.       It is not clear why higher doses of AD80 were used in anchorage-independent growth and 3D spheroid assay when it seems that lower doses will be adequate for inhibition.

6.       For combination with gemcitabine, the combination index should be calculated to check whether the effects are additive or synergistic.

Author Response

Reviewer #2

The article by Lima et al evaluated the efficacy of a multikinase inhibitor AD80 in pancreatic cancer in vitro. They had shown that AD80 lowered cell viability and clonogenicity and increased polyploidy in pancreatic cancer cells in vitro. They also performed Western blot and imaging studies to understand the molecular mechanism. Not surprisingly, AD80 showed alterations in multiple signaling pathways due to its role as a multikinase inhibitor. A targeted gene expression analysis by PCR array showed that AD80 could modulate a significant portion of autophagy-related genes and pathway analyses revealed its association with vacuole organization, macroautophagy, response to starvation, and cellular response to extracellular stimulus among others. Finally, the efficacy of AD80 in inhibiting growth in anchorage-independent soft agar assay as well as in 3D spheroid models was determined, albeit at higher dose levels than previous experiments. However, the manuscript needs significant improvement.

1. First and foremost, no in vivo xenograft studies were performed to determine the efficacy of AD80 as a viable therapeutic in pancreatic cancer. Multikinase inhibitors typically show very good in vitro efficacy due to their multi-target inhibition, however, they might cause significant toxicity at therapeutic doses in vivo due to the same reason. Hence, an in vivo experiment is needed to nullify any potential toxicity issues. This is a serious limitation of this study.

Authors' response: The authors thank the reviewer for the valuable suggestion and the opportunity to clarify this topic. Our research group is initiating studies in pancreatic cancer, and unfortunately, we do not have standardized murine models for pancreatic cancer in our laboratory, which will take some time. However, the most significant limitation at the moment is the lack of research funding needed to purchase animals for in vivo studies of AD80. We hope that the reviewer understands our difficulties at the moment. We acknowledge this limitation in the discussion of the revised version of the manuscript.

The following sentences have been included in the revised version of the manuscript:

"Our study opens perspectives for novel antineoplastic agents in pancreatic cancer; however, it has limitations. The scope of the present study was based on in vitro studies using pancreatic cancer cell lines, which provided cellular and molecular information. Investigating the effects in animal models of pancreatic cancer (e.g., ectopic or orthotropic tumor models) is needed to define the selectivity and toxicity of the drug. The results could lead to clinical trials."

2. The effect of AD80 on non-cancer cells such as HPDE cells should also be tested.

Authors' response: The authors thank the reviewer for the suggestion and opportunity to improve the presentation of our results. Following the reviewer's suggestion, we have added cell viability experiments in HPDE cells (non-tumor pancreatic cells) in the revised version of the manuscript.

The following sentence has been included in the revised version of the manuscript:

"In HPDE cells, IC50 values ranged from 1.48 to 4.78 µM and showed a favorable selectivity index (SI > 1) for MIA PaCa-2 and AsPC-1 cells after the 24 and 48 h treatments (Supplementary Figure 2)."

3. Please include cell cycle checkpoint markers ATM/CHK2, and ATR/CHK1 in western blot experiments. This is important since mitotic catastrophe is typically caused by the deregulation of cell cycle checkpoints.

Authors' response: Following the reviewer's suggestion, we added expression analysis for p-CHK1/CHK1 and p-CHK2/CHK2. Additionally, we include a panel of genes triggered in response to DNA damage and participating in cell cycle checkpoints, including CDKN1A, CDKN1B, GADD45A, BAX, BBC2, PMAIP1 and BCL2L11. These data are presented in the new Supplementary Figure 2.

The following sentences have been included in the revised version of the manuscript:

"Confirming the presence of mitotic aberrations and increased γHA2X, increased levels of CHK1 and/or CHK2 phosphorylation, as well as induction of genes involved in response to DNA damage, were observed in pancreatic cells upon AD80 exposure (Supplementary Figure 3)."

"Quantitative PCR

Total RNA from MIA PaCa-2, PANC-1 and AsPC-1 cells treated with vehicle or AD80 (1 μM) for 72 h was obtained using TRIzol reagent (Thermo Fisher Scientific, Inc.). cDNA was synthesized from 1 µg RNA using a High-Capacity cDNA Reverse Transcription Kit (Thermo Fisher Scientific, Inc.). Quantitative PCR (qPCR) was performed using a QuantStudio 3 Real-Time PCR System (Thermo Fisher Scientific, Inc.), a SybrGreen System, and specific primers (Supplementary Table 1). HPRT1 and ACTB were used as reference genes. Relative quantification values were calculated using the 2-ΔΔCT equation [15]. A negative 'No Template Control' was included for each primer pair.”

4. While p62 is considered an autophagic flux marker, alternative experiments like comparing LC3B conversion in the presence and absence of a lysosomal inhibitor such as bafilomycin-A1 should be performed to confirm the flux.

Authors' response: The authors thank the reviewer for the opportunity to incorporate the suggested experiment and improve the interpretation of our results. Following the reviewer's suggestion, we added an experiment comparing LC3B conversion in the presence and absence of a lysosomal inhibitor, bafilomycin A1, to confirm the effects of AD80 in the autophagic flux.

The following sentences have been included in the revised version of the manuscript:

"In PANC-1 cells, treatment with bafilomycin A1, a potent autophagic flux inhibitor, prevented the consumption of LC3B and SQSTM1 triggered by AD80, corroborating the hypothesis that AD80 is an autophagy inducer (Supplementary Figure 4)."

5. It is not clear why higher doses of AD80 were used in anchorage-independent growth and 3D spheroid assay when it seems that lower doses will be adequate for inhibition.

Authors' response: The choice of concentrations was made after a pilot study. It has been well established in the literature that more complex models, such as 3D, are more resistant to in vitro treatments. In addition, concentrations close to 1 µM (the same used in molecular assays) are included and showed promising results.

6. For combination with gemcitabine, the combination index should be calculated to check whether the effects are additive or synergistic.

Authors' response: There are many controversies in using the combinatorial index for antineoplastic agents, and these results can be misinterpreted depending on the experimental design. Thus, we adopted the concepts of pharmacology to describe our results: (i) if the effects observed in the combined treatments are greater than the individual effects, then the effects are potentiating; (ii) if the effects observed in the combined treatments are equal to the sum of the individual effects, then the effects are additive; (iii) if the observed effects of the combined treatments are greater than the sum of the individual effects, then the effects are synergistic. Our results indicate that the observed effects are potentiating.

Round 2

Reviewer 2 Report

The authors have addressed majority of the concerns raised during prior round of review. I have no more comments.